# Semiempirical Potential in Kinetics Calculations on the HC_3_N + CN Reaction

**DOI:** 10.3390/molecules27072297

**Published:** 2022-04-01

**Authors:** Emília Valença Ferreira de Aragão, Luca Mancini, Noelia Faginas-Lago, Marzio Rosi, Dimitrios Skouteris, Fernando Pirani

**Affiliations:** 1Master-Tec Srl, Via Sicilia 41, 06128 Perugia, Italy; emilia.dearagao@studenti.unipg.it (E.V.F.d.A.); d_skouteris@hotmail.it (D.S.); 2Dipartimento di Chimica, Biologia e Biotecnologie, Università degli Studi di Perugia, 06123 Perugia, Italy; luca.mancini2@studenti.unipg.it (L.M.); fernando.pirani@unipg.it (F.P.); 3Dipartimento di Ingegneria Civile ed Ambientale, Università degli Studi di Perugia, 06125 Perugia, Italy

**Keywords:** astrochemistry, ab initio calculations, semiempirical potential energy surface, Improved Lennard–Jones, chemical reactions, chemical calculations

## Abstract

The reaction between the cyano radical CN and cyanoacetylene molecule HC${}_{3}$N is of great interest in different astronomical fields, from star-forming regions to planetary atmospheres. In this work, we present a new synergistic theoretical approach for the derivation of the rate coefficient for gas phase neutral-neutral reactions. Statistic RRKM calculations on the Potential Energy Surface are coupled with a semiempirical analysis of the initial bimolecular interaction. The value of the rate coefficient for the HC${}_{3}$N + CN → H + NCCCCN reaction obtained with this method is compared with previous theoretical and experimental investigations, showing strengths and weaknesses of the new presented approach.

## 1. Introduction

Since the detection of the first molecular species, namely, the CH [1] and CN [2,3] molecules in dark clouds as well as in the Orion nebula and in the W51 region, huge efforts have been performed in order to unveil molecular complexity and understand the chemistry of the interstellar medium (ISM). Currently, more than 241 species have been detected in different regions of space [4], starting from simple molecules, up to more complex species with a strong prebiotic potential, such as formamide [5]. The understanding of the chemical processes which are considered the responsible for the formation and destruction of interstellar molecules in harsh conditions, where the temperature can go down to 10 K and the particle density can be as low as 10${}^{4}$ particles cm${}^{-3}$ [6], is a challenge for modern astrochemists. Despite different experimental techniques have been developed for this purpose [7,8], a theoretical approach appears to be fundamental for a global understanding of the reaction mechanisms.

In the last few years, a well-established computational strategy has been adopted to characterize all basic effects controlling the stereodynamics of gas phase reactions relevant in the astrochemical field [9,10,11]. A first DFT (Density Functional Theory) investigation is performed, through the use of hybrid functionals, for the calculation of geometrical structures and vibrational frequencies for all the stationary points included in the Potential Energy Surface (PES), namely, reactants, possible products, intermediates and transition states. Subsequently, the coupled-cluster CCSD(T) method is used to obtain more accurate values of energy to include in the PES. The data coming from electronic structure calculations are then used to perform a statistical analysis, through the use of capture theory and RRKM theory [12], in order to obtain reliable values for rate constants and branching ratios. In this work we report an additional integrated approach for the kinetic analysis of the reaction between the CN radical and the cyanoacetylene HC${}_{3}$N molecule, by employing a semiempirical method for the analysis of the initial bimolecular approach. In details, the Improved Lennard–Jones (ILJ) [13] potential is used in order to obtain a better description of the long-range interaction within the intermolecular potential. The ILJ potential can be considered as a refinement of the classical Lennard–Jones potential, allowing to obtain a better reproduction of both the long-range attraction and the short-range repulsion with a simple formulation.

The reaction between CN and cyanoacetylene is of great interest in different astronomical fields, including interstellar environments and planetary atmospheres. The CN radical is one of the first detected molecules in space [2,3], through the observation of spectral emission attributed to the rotational transition J = 1 → J = 0 both in the Orion Nebula and in the W51 region, while cyanoacetylene, HC${}_{3}$N, is among the first observed organic molecules, through the detection of the J = 0 → 1 transition in the galactic radio source Sgr B2 [14]. From that moment the HC${}_{3}$N molecule has been detected in a huge variety of astronomical environments, including dark molecular clouds, e.g., TMC-1 [15,16,17], low-mass protostars [18], protoplanetary disks [19,20], the carbon-rich star IRC+10216 [21] and planetary atmospheres [22,23]. The cyanopolyynes, a group of molecules sharing the same characteristics, namely, a linear structure with alternating carbon-carbon single and triple bonds characterized by the presence of a cyano group are ubiquitous in the ISM. Different molecules belonging to the family of cyanopolyynes have been detected in space, with increasing number of carbon atoms [24,25]. A currently accepted strategy for the synthesis of cyanopolyynes in the ISM starts from the collision of the CN radical with unsaturated hydrocarbons [26], including acetylene, leading to the formation of cyanoacetylene. Successive chain-elongation reactions involving cyanoacetylene and the C${}_{2}$H radical [27] can be the responsible for the formation of more complex cyanopolyynes. Nevertheless, a different fate for the new formed cyanoacetylene is the collision with another CN radical, leading to the formation of dicyanoacetylene (NCCCCN) and terminating the chain-elongation process. Therefore, the reaction with cyano radical is considered as a chain-termination reaction. Since the dicyanoacetylene molecule cannot be detected in the ISM due to the lack of permanent electric dipole moment, chemical information about the rate constant of the reaction between CN and cyanoacetylene appears to be pivotal. An additional astronomical environment in which the reactions between various N-containing molecules play a key role is related to planetary atmospheres. In particular, one of the most intriguing bodies in our solar system appears to be Titan, the largest satellite of Saturn, as well as one of the only body in our Solar System to have a thick atmosphere mainly containing molecular nitrogen, like Earth. Together with nitrogen, several hydrocarbons have been detected through the atmosphere of Titan, including cyanoacetylene HC${}_{3}$N, which appears to be one of the most abundant nitriles on Titan [22,23]. Furthermore, this molecular species is believed to be an important precursor of more complex N-bearing species in Titan’s organic aerosols [28,29]. An accurate analysis of the reaction between CN and HC${}_{3}$N, to be include in photochemical models, can help clarifying the fate of cyanopolyynes on Titan’s atmosphere.

The title reaction has already been analysed through different investigations in the last few years. In 1987, Yung et al. considered the reaction between the cyano radical and HC${}_{3}$N as a possible route for the formation of C${}_{4}$N${}_{2}$ (dicyanoacetylene) in the atmosphere of Titan [30]. The first experimental investigation of the rate coefficient was then performed in 1989 by Halpern et al. [31], obtaining a rate coefficient of (1.7 ± 0.08) × 10${}^{-11}$ cm${}^{3}$·s${}^{-1}$ at room temperature. A later work by Faure et al. [32] allowed to predict the canonical rate coefficient through the use of a semiempirical model in a temperature range from 10 to 295 K. The resulting fitting [33,34,35] led to a rate coefficient expressed as: k(T) = 1.85 × 10${}^{-11}$ (T/300)${}^{1.93}$ exp(−34.5/T) cm${}^{3}$·s${}^{-1}$. A new experimental analysis has been performed in 2013 [36] using both pulsed laser photolysis-laser-induced fluorescence and the CRESU (Reaction Kinetics in Uniform Supersonic Flow) technique, allowing to evaluate the rate coefficients at very low temperatures. The measured rate coefficients were represented by the expression k(T) = 1.79 × 10${}^{-11}$ (T/300)${}^{-0.67}$ cm${}^{3}$·s${}^{-1}$, with a root-mean squared error of 0.61 × 10${}^{-11}$ cm${}^{3}$·s${}^{-1}$ and an estimated systematic error of 10%. In conjunction with the experiment, the authors presented a theoretical analysis of the rate coefficient employing a two transition state (2TS) model [37], with k(T) = [1.97 × 10${}^{-8}$ T${}^{-1.51}$ exp(−3.24/T) + 4.85 × 10${}^{-13}$ T${}^{0.563}$ exp(17.6/T)] cm${}^{3}$·s${}^{-1}$ for temperatures ranging from 5 to 400 K. A key point for the correct estimation of the reaction rate constant is the analysis of the initial bimolecular approach. In the present work we report a semi-empirical investigation of the initial bimolecular approach between the cyanoacetylene molecule and cyano radical. The semi empirical potential related to the long-range interaction, evaluated at different angles, which allowed a better comprehension of the first steps of the reaction, has been included in the statistical RRKM calculations in order to derive the final rate constant of the reaction. The main purpose of this work is then to present a methodology, which does not intend replace more quantitative treatments, useful to provide in a simple way the magnitude order of rate coefficients values, for these and many other elementary reactions when occurring at low temperature, and to cast light on the microscopic reaction mechanisms. In particular, it is demonstrated that the use of RRKM methodology suggests that low temperature rate coefficients are mainly determined by the critical balance of capture effects by long range forces and back dissociation events. A comparison with results of other treatments is also presented and discussed.

## 2. Results

The theoretical investigation of the PES for the HC${}_{3}$N + CN reaction revealed only one exothermic channel, related to the formation of dicyanoacetylene C${}_{4}$N${}_{2}$ [38]. In details, the reaction starts with the formation of a first long-range complex, named ‘vdW’, located 4.1 kJ·mol${}^{-1}$ below the reactant energy asymptote. A transition state, ‘TS-vdW-INT’, must be overcome in order to form an intermediate, ‘INT’, characterized by the formation of a new C-C bond, with a relative energy of −231.9 kJ·mol${}^{-1}$. Subsequently the products C${}_{4}$N${}_{2}$ + H can be formed through a barrier of 189.2 kJ·mol${}^{-1}$, represented by the transition state ‘TS-INT-Product’, which clearly shows the breaking of the C-H bond. The energies of all the stationary points, including reactant and products, evaluated at the CCSD(T)/aug-cc-pVTZ//M06-2X/6-311+G(d,p) level of theory, are reported in Table 1.

The intermolecular potential $V_{Entrance}$ (cf. Equation (Equation 1)) has been semi-empirically computed by building a series of structures based on the vdW geometry. The two optimized reactants were placed in a manner where $\hat{C_{Rad}C_{1}C_{2}}$ is 90° and for different values of $\hat{C_{1}C_{Rad}N_{Rad}}$, such as 120°, 135°, 150° and 180° (Figure 1). Then the intermolecular potential was calculated for each geometry for a range of C${}_{1}$-C${}_{Rad}$ distances from 2.5 up to 12.0 Å. The resulting curves of the intermolecular potential for all values of $\hat{C_{1}C_{Rad}N_{Rad}}$ are displayed in Figure 2. The choice to represent the C${}_{1}$-C${}_{Rad}$ distance in the *x*-axis rather than the distance between center-of-mass of each molecule is justified by the fact that a bond is forming between these two atoms. For distances shorter than 3.5 Å, repulsion causes the energy value to increase exponentially with the decreasing C${}_{1}$-C${}_{Rad}$ distance. In all curves, the bottom of the well is located at about 3.6 Å, while in the vdW geometry, the C${}_{1}$-C${}_{Rad}$ distance is 2.74 Å. The energy in this region is −3.79 kJ·mol${}^{-1}$, −2.77 kJ·mol${}^{-1}$, −1.95 kJ·mol${}^{-1}$ and −0.81 kJ·mol${}^{-1}$ for $\hat{C_{1}C_{Rad}N_{Rad}}$ angle values of 120°, 135°, 150° and 180°, respectively. In comparison, according to the results of electronic structure calculations, the relative energy attributed to the vdW structure is −4.1 kJ·mol${}^{-1}$. With the increasing distance, the intermolecular potential energy at the configuration $\hat{C_{1}C_{Rad}N_{Rad}}$ = 180° becomes positive, which indicates that the approach of fragments at that angle is hindered. In contrast, the intermolecular potential curves of the configurations $\hat{C_{1}C_{Rad}N_{Rad}}$ = [120°; 135°; 150°] stays below 0 kJ·mol${}^{-1}$, reaching the asymptotic value at long-range. This behavior is expected from variational calculations employing ab initio methods. However, to build a potential energy curve from ab initio variational calculations, every point of that curve must correspond to an optimized geometry. In particular, the calculation is performed by freezing one coordinate (here the C${}_{1}$-C${}_{Rad}$ distance) and optimizing the rest of the interatomic distances. After a number of calculation cycles, the system might converge to an optimized geometry that correspond to the reaction pathway, or not. In the latter case, the particular point cannot be used to build the variational curve. Alternative cost-effective methods have to be considered to simulate the approach between the two reactants. In comparison, the intermolecular potential approach allows the computation of the potential values instantaneously for a large number of points, within a large interatomic distance range. Therefore, we have used the values of the potential between 4.0 Å and 12.0 Å to perform a linear fitting and evaluate the asymptotic attraction coefficient C${}_{6}$ (cf. Equation (Equation 5)). Such coefficient determines the capture efficiency by long-range forces that lead to the formation, by collision of reagents, of the weakly bound precursor state, controlling the stereo-dynamical evolution of the system especially under subthermal conditions. Values of C${}_{6}$ coefficient, obtained for different approach geometries, are displayed in Table 2.

With that information and exploiting the guidelines reported in [12], it was possible to estimate the rate of each unimolecular process of the reactive channel, i.e., the conversion from vdW to INT, the reverse process from INT to vdW and the formation of C${}_{4}$N${}_{2}$ + H from INT, by employing RRKM theory. In addition, the back-dissociation from the van der Waals adduct to the reactants was also computed by multiplying the capture rate coefficient with the ratio between density of states of vdW and the density of states of the reactants. All these four processes were taken into account in the solving of the master equation, allowing us to obtain the rate coefficient k(E) of the formation of dicyanoacetylene. From that, the canonical rate coefficient k(T) was derived for a temperature range spanning from 1 to 300 K.

In Figure 3, we see the reaction rate coefficients spanning from 1 K to 300 K for different values of $\hat{C_{1}C_{Rad}N_{Rad}}$. We observe that for all angles of attack, the rate coefficients are in the same order of magnitude, and the values starts to coincide around 10 K. The rate coefficients were found to decrease in the 10–70 K temperature range, which translates the effect of the back-dissociation of vdW increasing with the temperature at a faster rate than the product formation. For temperatures higher than 70 K, the rate coefficients increase monotonically. Taking into account this difference in behavior, we found more relevant to fit separately rate coefficients between 10 and 70 K and rate coefficients spanning from 71 to 300 K. In both intervals, the models seems to reproduce better the behavior of the data than the fitting performed for the range of 10 to 300 K (see Figure 4). We report the optimal values for alpha, beta and gamma in Table 3.

## 3. Discussion

According to the analysis performed through electronic structure calculations, only one exothermic product can be observed, dicyanoacetylene. In the ISM context, this is also the only relevant product, since the conditions of temperature and particle density allow only barrierless exothermic processes to take place. The results of the electronic structure calculations also echoes previous results published in [40], with some differences in the energy values of the stationary points, which can be attributed to the use of different basis-sets. Figure 5 shows a comparison between the values of the rate coefficients estimated in this study and from previous publications: [31,32,36]. In the earliest paper, the authors estimated that at room temperature, the rate coefficient of the HC${}_{3}$N + CN reaction is 1.70 ± 0.08 × 10${}^{-11}$ cm${}^{3}$molecule${}^{-1}$s${}^{-1}$. The value obtained by our model at 300 K is 2.90 × 10${}^{-11}$ cm${}^{3}$molecule${}^{-1}$s${}^{-1}$, so it is at the same order of magnitude and a slight overestimation by a factor or 1.6 from the experimental rate coefficient. In the work from 2009, Faure et al. estimated the rate coefficient employing another semiempirical capture model. According to the authors, their proposed model exploits the long range dispersion attraction C${}_{6}$ coefficient estimated by means the London formula. The C${}_{6}$ coefficient is subsequently employed in the calculation of the capture rate constant. At low temperature, the rate coefficient is of the order of magnitude of 10${}^{-10}$ cm${}^{3}$molecule${}^{-1}$s${}^{-1}$, which appears to be high for neutral-neutral reactions and different from the experimental values. In our model, we predict rate coefficients from 10 to 100 times smaller.

Finally, we compare our model to Cheikh Sid Ely et al. work from 2013 [36]. As of this date, the rate coefficients derived from measurements coupling pulsed laser photolysis-laser-induced fluorescence and the CRESU technique are the most complete set of experimental data. Our model and the CRESU measurements show better agreement at room temperature. At very low temperatures, where the data disagree the most, our model underestimates the values of the rate coefficient by a factor of 5 (22 K). Nevertheless, the values are in the same order of magnitude (10${}^{-11}$ cm${}^{3}$molecule${}^{-1}$s${}^{-1}$). In terms of trend, in both models we see a decrease of the rate coefficient for temperatures ranging from 20 to 70 K. For temperatures above 70 K, the experimental data continues to show a decrease, while our model shows an increase of the rate coefficient.

In the same work, the authors have computed theoretical rate coefficients by means of the 2TS model. They have combined ZPE corrections at CCSD(T)/cc-pVTZ level, complete basis-set (CBS) extrapolation, core-valence and relativistic corrections to calculate the energy of the initial saddle point with high accuracy. The resulting theoretical rate coefficient was lower than the experimental one, nevertheless a better agreement can be noticed by increasing the level of calculations. In comparison, our model employs DFT method to compute all stationary points in the reactive channel, with the advantage of a less expensive procedure, which can be applied quickly to other systems. The data obtained in the present work lead to lower values of the rate coefficient. This can be due to the fact that the transition state obtained at the CCSD(T)/aug-cc-pVTZ//M06-2X/6-311+G(d,p) level of theory, appears to be higher in energy than the value obtained in [36], leading to a higher barrier to overcome in order to form the intermediate ‘INT’. Moreover, the calculation of our theoretical rate coefficient takes into account all stationary points in the reactive channel, in particular the van der Waals adduct. In this way a step to calculate the back-dissociation was included. The results have shown that for temperatures higher than 10 K, the back-dissociation rate constant is of the order of magnitude of 10${}^{-10}$ cm${}^{3}$molecule${}^{-1}$s${}^{-1}$. For temperatures above 20 K, the back-dissociation branching ratio is 90%, which shows that most of the collisions do not lead to the formation of dicyanoacetylene. The method presented in this work is not able to properly describe on quantitative ground the global trend of the rate coefficient, even though it appears to be a valid alternative for the estimation of the capture effects by long range forces, basically controlled the effective C${}_{6}$ attraction coefficient, decreasing the computational cost. A further refinement of the method can be considered in order to obtain results more in line with the experimental analysis.

## 4. Theoretical Methods

As previously mentioned, the HC${}_{3}$N + CN reaction has been analysed adopting a well-established computational strategy, already used successfully in several cases [41,42,43,44,45,46,47]. Based on the recent work on the doublet potential energy surface by the authors of this paper [38], the stationary points along the channel leading to dicyanoacetylene have been characterized by performing density functional theory calculations with the M06-2X [48] hybrid functional in conjunction with the 6-311+G(d,p) basis set [49,50]. The same M06-2X/6-311+G(d,p) level of theory has been employed to perform harmonic vibrational frequencies calculations, in order to determine the nature of each stationary point: minimum if all the frequencies are real and transition state if there is one, and only one, imaginary frequency. Saddle points have been assigned through Intrinsic Reaction Coordinates (IRC) [51,52] calculations. Successively, the more accurate coupled cluster theory, including single and double excitations as well as a perturbative estimate of connected triples [CCSD(T)] [53,54,55], has been used in conjunction with the correlation consistent valence polarised set aug-cc-pVTZ [56] for the calculations of the energy of each stationary point. The so obtained CCSD(T) energies have been corrected by including the zero-point energy correction, computed using the scaled harmonic vibrational frequencies obtained at the M06-2X/6-311+G(d,p) level of theory. All calculations have been carried out using the GAUSSIAN 09 [57] software.

The kinetics of the neutral-neutral reaction HC${}_{3}$N + CN have been investigated using a combination of capture and RRKM theories following a well-established protocol employed in previous works of our group [58,59,60]. Initially, the approach between cyano radical and cyanoacetylene is represented using a semiempirical model. Assuming the intermolecular interaction energy as the sum of electrostatic and non-electrostatic contributions, i.e.,
(1)$$V_{Entrance}=V_{\mathrm{electrostatic}}+V_{\mathrm{non-electrostatic}}$$

The electrostatic component $V_{electrostatic}$ is defined by Coulomb’s law:(2)$$V_{\mathrm{electrostatic}}\left(r\right)=\frac{1}{4\pi \varepsilon_{0}}\sum_{i=1}^{5}\sum_{j=1}^{2}\frac{q_{i}q_{j}}{r_{ij}}$$
where $\varepsilon_{0}$ is the vacuum permittivity, $q_{i}$ corresponds to the partial charge on an atom of the cyanoacetylene molecule, $q_{j}$ is the partial charge on an atom of the cyano radical and $r_{ij}$ is the distance between the two atoms involved, obtained deconvolving r into partial components. The ESP partial charges for each atom, used in order to derive the electrostatic term have been computed at M06-2X/6-311+G(d,p) level of theory (cf. Table 4).

The non-electrostatic component of Equation (Equation 1) has been represented within the pair-wise additivity approach of the interaction. In particular, $V_{non-electrostatic}$ has been calculated as the sum of several contribution, one for each pair of atoms, expressed through the Improved Lennard–Jones functions [13,61,62], given by the formula
(3)$$V_{ij}\left(r\right)=\varepsilon_{ij}\left[\frac{m}{n(r_{ij})-m}{\left(\frac{r_{m_{ij}}}{r_{ij}}\right)}^{n(r_{ij})}-\frac{n(r_{ij})}{n(r_{ij})-m}{\left(\frac{r_{m_{ij}}}{r_{ij}}\right)}^{m}\right]$$
where, $\varepsilon_{ij}$ and $r_{m_{ij}}$ are, respectively, the potential well depth and the equilibrium distance associated to the considered *ij* pair, and $r_{ij}$ is the distance between the *i* and *j* atoms located on the two interacting partners, obtained again by the r decomposition. Note that from the interaction point of view, each atom, involved in the formation of chemical bonds within the same molecular framework, must be considered as an effective atom, that is having an effective polarizability value rather different respect to that of the same atom isolated in gas phase. However, the sum of the effective components must be consistent with the free radical/molecular polarizability value. The parameter $n(r_{ij})$ was first introduced in a work by [62] as a modification of the Maitland-Smith correction to the Lennard–Jones model. According to [13,62], *m* assumes the value of 1 for ion-ion interaction, 2 for ion-permanent dipole, 4 for ion-induced dipole, and 6 for neutral-neutral systems, as the present one. To modulate the decline of the repulsion and the strength of the attraction in Equation (Equation 3), $n(r_{ij})$ takes the following form:(4)$$n\left(r_{ij}\right)=\beta +4{\left(\frac{r_{ij}}{r_{m_{ij}}}\right)}^{2}$$
where $\frac{r_{ij}}{r_{m_{ij}}}$ assumes the meaning of a reduced distance and β is a parameter related to the nature and the hardness of the interacting particles [62,63,64,65]. In the current system, the value assigned to β is 7. The values of the parameters $\varepsilon_{ij}$ and $r_{m_{ij}}$ introduced in the Improved Lennard–Jones formulation are reported in Table 5.

After calculating the intermolecular interaction energy, the intermolecular potential $V_{Entrance}$ at long-range assumes the form of
(5)$$V_{Entrance}=-\frac{C_{6}}{r^{6}}+b$$
where *r* is the separation distance between the two interacting partners and C${}_{6}$ represents here the global (effective) attraction coefficient, determined by the critical balance dispersion and electrostatic effects, whose value depends on the approach geometry of the reactants. Moreover, b is a parameter which empirically accounts for the different radial dependence of the two considered long range interaction components. In this work, *r* represents the C${}_{1}$C${}_{Rad}$ distance and the estimated C${}_{6}$ value ranges from 0.7 to 4.7 E${}_{h}\;\cdot \;{Å}^{6}$ for different angles of attack (see Table 2). The obtained C${}_{6}$ coefficient was employed in the calculation of the capture cross-section [58]:(6)$$\sigma \left(E\right)=\pi \times 3\times 2^{-\frac{2}{3}}\times {\left(\frac{C_{6}}{E}\right)}^{\frac{1}{3}}$$
where E is the translational energy. By multiplying the capture cross section by the collision velocity, namely, ${\left(2E/\mu \right)}^{\frac{1}{2}}$ with μ representing the reduced mass of the reactants, the capture rate coefficient can be obtained [58]. Subsequently, the rate coefficient of each unimolecular step at a specific energy E has been calculated taking into account the vibrational frequencies of the intermediate and of the saddle point structures. The following formula was employed:(7)$$k\left(E\right)=\frac{N(E)}{h\rho (E)}$$
where *N(E)* is the sum of states of a saddle point at a specific energy *E*, *h* is the Planck constant and ρ(*E*) is the density of states of an intermediate. With the capture rate constant and the unimolecular rate constants, it was possible to solve the master equation for the system and consequently derive rate constants for the overall reaction in function of the energy. In the master equation treatment, it was assumed that the total energy remains constant throughout the whole reaction scheme, i.e., up until this step the system was being treated considering the microcanonical ensemble. Afterwards, the rate coefficient as a function of the temperature has been obtained by carrying out a Boltzmann averaging with respect to the reactants population. Finally, the data were fitted to the Arrhenius-Kooij [33] 3-parameters equation for a temperature range from 10 to 300 K:(8)$$k\left(T\right)=\alpha {\left(\frac{T}{300}\right)}^{\beta_{T}}e^{-\frac{\gamma }{T}}$$

This equation is currently used in a variety of astrochemical models [34,35] to express the rate coefficient of bimolecular reactions.

## 5. Conclusions

In this work, we present a new theoretical study of the reaction between cyanoacetylene and cyano radical, adopting a synergistic approach between a semiempirical formulation for the long-range interaction potential and rigorous ab initio calculations, coupled with a statistic analysis. Based on the results of the electronic structure calculations, the reaction involving cyanoacetylene and cyano radical leads to only one exothermic product, related to the formation of dicyanoacetylene, which appears to be the only possible product formed considering the harsh condition of the intertsellar medium. The theoretical rate coefficient related to the formation of the exothermic product was estimated by combining Capture theory and RRKM theory. The rate coefficient was fit to k(T) = 5.4 × 10${}^{-12}$ (T/300)${}^{-0.2}$ exp${}^{18/T}$ for a temperature range of 10 to 70 K and to k(T) = 2.0 × 10${}^{-11}$ (T/300)${}^{1.6}$ exp${}^{120/T}$ from 71 to 300 K. The kinetic study also reveals that the back-dissociation process is dominant for temperatures higher than 20 K. Therefore, even when the van der Waals adduct is formed it is more probable that the system will reconvert into cyanoacetylene and cyano radical. The results obtained in this work are coherent with previous investigations of this systems by other groups. Despite some differences in the global trend of the overall rate constant, the new combined method presented for the calculation of the rate coefficient, combining semiempirical and statistic analysis, yielded similar results in comparison to more expensive methods. Besides being cost-effective, our protocol is transferable, therefore it will be applied to different systems in future studies.

## Figures and Tables

**Figure 1 molecules-27-02297-f001:**
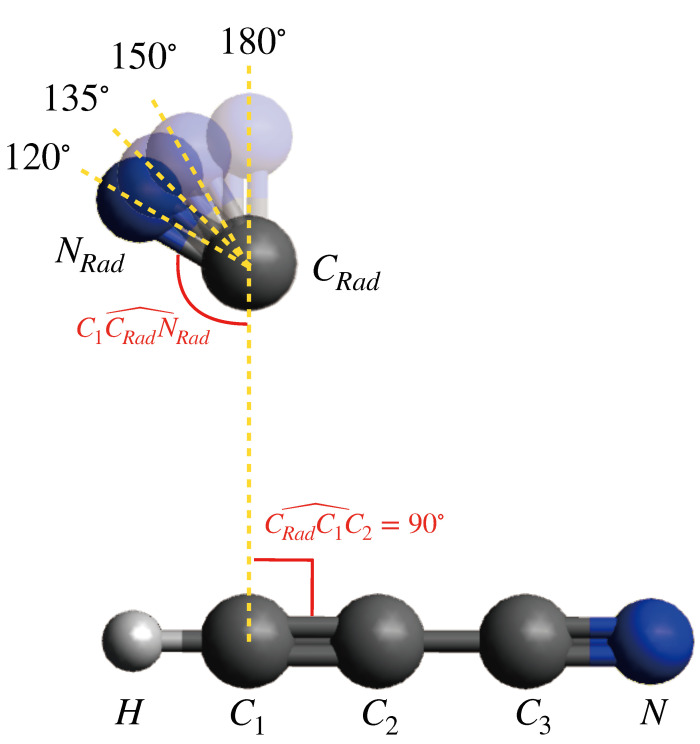
Scheme showing the relevant $\hat{C_{1}C_{Rad}N_{Rad}}$ angles (in degrees) that where fixed to represent the approach between the reactants.

**Figure 2 molecules-27-02297-f002:**
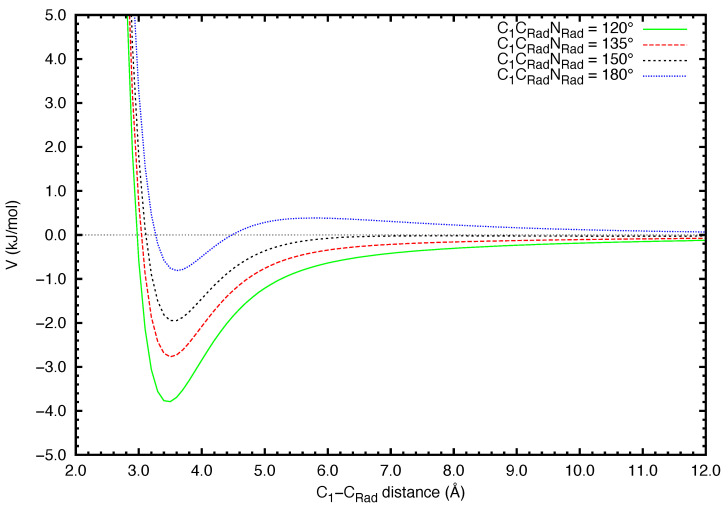
Curves of the intermolecular potential $V_{Entrance}$ (in kJ·mol${}^{-1}$) representing the initial bimolecular approach at different $\hat{C_{1}C_{Rad}N_{Rad}}$ angles (in degrees).

**Figure 3 molecules-27-02297-f003:**
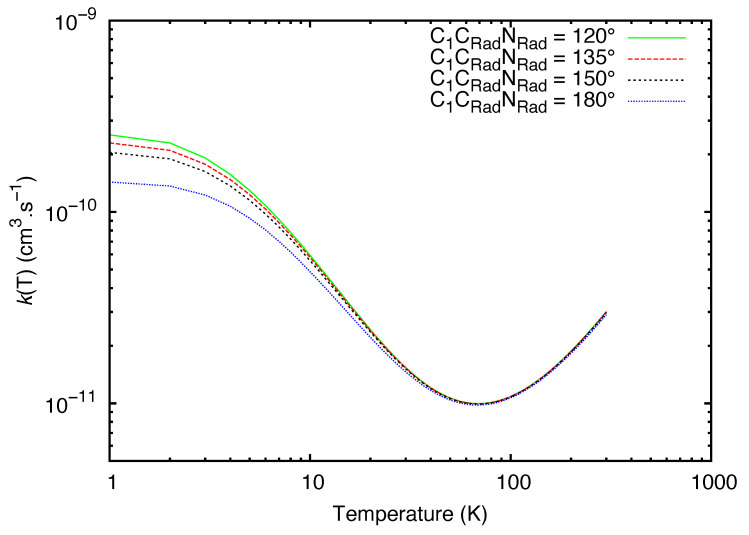
Comparison of the theoretical rate coefficients at different $\hat{C_{1}C_{Rad}N_{Rad}}$ angles (in degrees).

**Figure 4 molecules-27-02297-f004:**
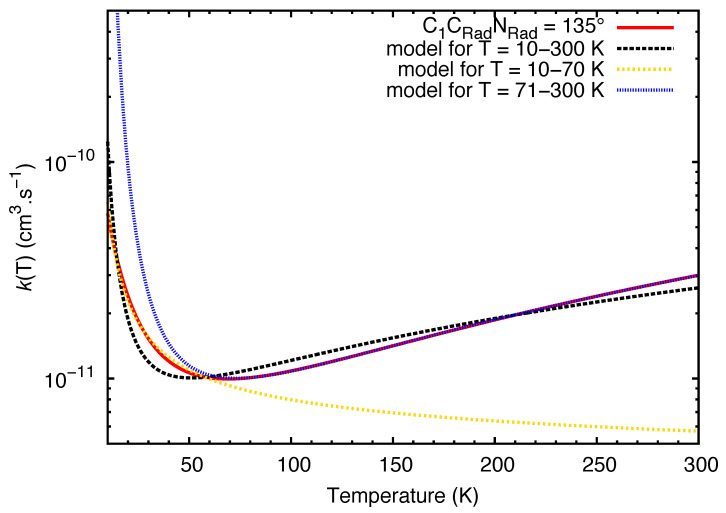
Comparison between fittings for $\hat{C_{1}C_{Rad}N_{Rad}}$ = 135${}^{\circ }$. The rate coefficients obtained through capture and RRKM theory are represented in red. In black, the fitting of the curve performed for the temperature range from 10 to 300 K. In yellow, the fitting of the data from 10 to 70 K. In blue, the fitting for a temperature spanning from 71 to 300 K.

**Figure 5 molecules-27-02297-f005:**
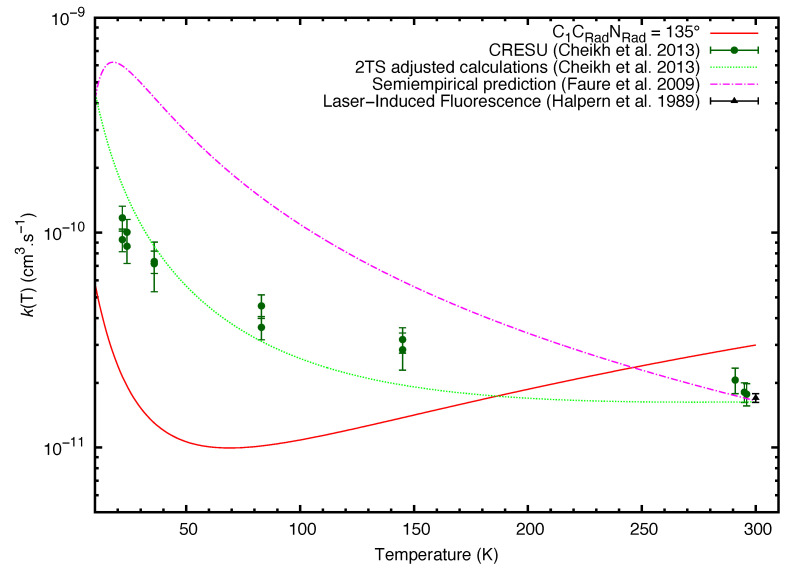
Rate coefficients for the reaction between cyanoacetylene and cyano radical (in cm${}^{3}$s${}^{-1}$) in function of the temperature (in K). In full-line red, data from this work at $\hat{C_{1}C_{Rad}N_{Rad}}$ = 135°. The room temperature rate coefficient evaluated in the work by Halpern et al. [31] is represented by a black triangle with error bars. The dashed-dotted magenta line, represents the semiempirical prediction by Faure et al. [32]. The dark-green circles with error bars represent the experimental rate coefficients and the green dotted line represents the theoretical rate coefficients, both extracted from the publication by Cheikh Sid Ely et al. [36].

**Table 1 molecules-27-02297-t001:** Computed energies a the CCSD(T)/aug-cc-pVTZ//M06-2X/6-311+G(d,p) level of theory.

	$E_{\mathrm{CCSD}(T)}$ ($E_{h}$)	${\mathrm{ZPE}}_{M06-2X}$ ($E_{h}$)	ΔE (kJ·mol${}^{-1}$)
HC${}_{3}$N + CN	−261.8584480	0.032858	0.0
vdW	−261.8604704	0.033320	−4.1
TS-vdW-INT	−261.8587898	0.033110	−0.2
INT	−261.9504299	0.036495	−231.9
TS-INT-Product	−261.8702081	0.028342	−42.7
C${}_{4}$N${}_{2}$ + H	−261.8795165	0.027620	−69.1

**Table 2 molecules-27-02297-t002:** Values of C${}_{6}$ and b of Equation (Equation 5). The linear fitting was carried out with values of $V_{Entrance}$ for $d_{C_{1}-C_{Rad}}$ = [4;12] Å. Note that 1 E${}_{h}$ = 2625.500 kJ·mol${}^{-1}$ [39].

$\hat{C_{1}C_{\mathrm{Rad}}N_{\mathrm{Rad}}}$	C${}_{6}$ (E${}_{h}\;\cdot \;$Å${}^{6}$)	b (E${}_{h}$)
120°	4.67	−9.60 × 10${}^{-5}$
135°	3.37	−4.66 × 10${}^{-5}$
150°	2.31	7.07 × 10${}^{-7}$
180°	0.73	8.64 × 10${}^{-5}$

**Table 3 molecules-27-02297-t003:** Values of α, $\beta_{T}$ and γ coefficients for the modified Arrhenius equation (Equation (Equation 8)).

Temperature Range (K)	$\hat{C_{1}C_{\mathrm{Rad}}N_{\mathrm{Rad}}}$	α (cm${}^{3}$·s${}^{-1}$)	$\beta_{T}$	γ
10−70	120°	5.37 × 10${}^{-12}$	−0.19	−18.80
	135°	5.37 × 10${}^{-12}$	−0.19	−18.47
	150°	5.36 × 10${}^{-12}$	−0.20	−17.97
	180°	5.41 × 10${}^{-12}$	−0.20	−16.39
71−300	120°	2.01 × 10${}^{-11}$	1.67	−121.77
	135°	2.00 × 10${}^{-11}$	1.66	−121.16
	150°	1.99 × 10${}^{-11}$	1.65	−120.31
	180°	1.96 × 10${}^{-11}$	1.62	−117.34

**Table 4 molecules-27-02297-t004:** Parameters for the electrostatic potential. ESP charges estimated at M06-2X/aug-cc-pVTZ level.

Atom	ESP Partial Charges (a.u.)
C${}_{1}$	−0.234
C${}_{2}$	0.120
C${}_{3}$	0.478
N	−0.424
H	0.300
C${}_{Rad}$	0.339
N${}_{Rad}$	−0.339

**Table 5 molecules-27-02297-t005:** Parameters for the non-electrostatic potential. Note that 1 meV = 0.09648534 kJ·mol${}^{-1}$ [39].

Interacting Pair	$r_{m_{\mathrm{ij}}}$ (Å)	$\varepsilon_{\mathrm{ij}}$ (meV)
C-C${}_{Rad}$	3.80	6.16
N-C${}_{Rad}$	3.73	6.07
H-C${}_{Rad}$	3.54	2.43
C-N${}_{Rad}$	3.71	5.95
N-N${}_{Rad}$	3.63	6.07
H-N${}_{Rad}$	3.41	2.55

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
