# Peer review of "Semiempirical Potential in Kinetics Calculations on the HC3N + CN Reaction"

_molecules, 2022, doi:10.3390/molecules27072297_

Round 1

Reviewer 1 Report

The paper "Semiempirical potential in kinetics calculations on the HC3N +
CN reaction" is devoted to the calculation of potential energy surface of the title reaction, intermolecular potential and evaluation of the dependenca of the rate constant on the temperature. The paper appears to be an interesting contribution to astrochemistry even though the applied approach fails (yet?) to satisfactorily describe the experimental data.

Some minor points:

  1. Attacking angle of 90o is metioned once; however, no data is given in either intermolecular potential or theoretical rate coefficients. Why not? And what would happen if angle is below 90o? The regularity on Fig. 2 suggests that the curve would drop even more (at angle <90o); however, it seems illogical as nitrogen would hinder the reaction being in between the interacting carbons.
  2.  The intermediate is so stable (-232 kJ mol-1); it makes me wonder why it even dissociates in either side of the reaction? Looks strange; usually, intermediate is unstable.
  3. The division of temperature range in two parts (1 to 70 and 71 to 300 K) seems not completely justified to me. I believe, you should show more clearly that this is necessary (one curve do not fit well enough to the calculated dependencies of k(T); and/or the division point should be exactly 70 K and not, e.g. 69.38 K or 72.21 K etc.).
  4. Why did you use ESP charges? Why not, e.g., Mulliken or Hirshfeld?

Reviewer 2 Report

In this manuscript, reaction of HC3N with CN radical was investigated using computational chemistry methods and the derived potential energy surface was used to obtain rate constants for the title reaction using statistical rate theory. The potential energy curve of the entrance channel had been described by Improved Lennard-Jones potential to obtain capture cross-section and then the capture rate coefficient. The unimolecular steps involving tight transition state are described by the statistical RRKM theory coupled with master equation (ME) analysis. This description is supposed to be an alternative to the Klippenstein’s two transition state theory (2TS) which was applied to the same system in 2013 (reference 36). Personally, I cannot see any superior achievements compared to the previous kinetic paper, therefore, I cannot recommend this paper to be published in Molecules. Additional major issues are as follows:

  • Only one of the possible reaction channels had been investigated. H-abstraction and other CN-addition can also be possible competitive channels. It should have been verified other channels are only minor at the temperature range of interest.
  • As it told in the Introduction, the particle density can be as low as 104 particle cm-3 which can cause at least two difficulties for the kinetic description of the title reaction, namely non-statistical behavior, and low probability of collision.
  • The values for C6 parameter were obtained from linear fit. This step is not obvious, and it is not well documented in the article. I am also wondering why it is necessary to use linear function for the fit when non-linear fitting is also possible.
  • ME usually requires collision parameter which is not given in the manuscript. Furthermore, the ME assumes energy distribution (e.g. Boltzmann) of the intermediate (INT). More detailed discussion of the applied kinetic theories is mandatory to provide insights why this kinetic model can be accepted for such complicated situation.
  • It is written in line 131 that not all the optimizations were converged. It is not clear whether this term refers to the wavefunction (DFT, HF or CCSD(T)) or the geometry. In both case, there are techniques which can be used to eliminate them entirely (e.g. Vshift keyword, smaller step size for the scan can also affect the convergence of the consecutive geometry optimization).

Reviewer 3 Report

The authors computed the rate coefficient for the HC3N + CN → H + NCCCCN
reaction and compared it with previous theoretical and experimental investigations. Before publishing it needs to address the following question.

1- It seems that the method (M0f-2X) used for calculations of ZPE is not enough good, which may have some effect on the rate constant, especially at large temperatures.

2- The calculated rate constant for low collision energy is quite good due to the long-range potential contributions but it fails after 100 K. An explanation is needed if this is due to the calculations method of rates or the method used for the PES. 

Round 2

Reviewer 2 Report

The authors claims that this manuscript is not an alternative to 2TS theory, but it is to ‘provide in a simple way the magnitude order of rate coefficients values’. If so, simple, well-established competitor theory, that is the conventional Transition State Theory (TST), can also be applied here for ‘TS-vdW-INT’. Provide a direct comparison.
